# Lifestyle and mood correlates of cardiometabolic risk in people with serious mental illness on second-generation antipsychotic medications

**Susanne U. Miedlich**[1]*, **Priya Sahay**[2], **Telva E. Olivares**[3], **J. Steven Lamberti**[3], **Diane S. Morse**[3,4], **Kevin P. Brazill**[3], **Kavaljit H. Chhabra**[5], **Lauren Bainbridge**[1]

1 Division of Endocrinology and Metabolism, Department of Medicine, University of Rochester School of Medicine and Dentistry, Rochester, New York, United States of America, 2 Barnegat Medical Associates, Toms River, New Jersey, United States of America, 3 Department of Psychiatry, University of Rochester School of Medicine and Dentistry, Rochester, New York, United States of America, 4 Department of Medicine, University of Rochester School of Medicine and Dentistry, Rochester, New York, United States of America, 5 Department of Pharmacology and Nutritional Sciences, College of Medicine, University of Kentucky, Lexington, KY, United States of America

* susanne_miedlich@urmc.rochester.edu

**Data Availability Statement:** All relevant data are within the manuscript and its Supporting Information files.

## Abstract

### Introduction

Cardiovascular morbidity and mortality are high in people with serious mental illness (SMI). This problem is mediated, at least in part, by metabolic side effects of second-generation antipsychotics (SGAs) and by unhealthy lifestyle behaviors. We asked whether oral glucose tolerance testing (oGTT) or hemoglobin A1c (HbA1c) is superior in identifying people with SMI at high cardiometabolic risk and whether this risk is shaped by mood, cognition, or life-style habits.

### Methods

We evaluated 40 patients with schizophrenia, schizoaffective, or bipolar disorder receiving SGAs by oGTT, HbA1c, comprehensive metabolic and lipid panels, and CRP. Mood was assessed using the Patient Health Questionnaire (PHQ-9), and cognition was assessed using the Saint Louis University Mental Status examination. Diet was assessed using the UK Diabetes and Diet Questionnaire (UKDDQ), and physical activity was assessed using daily step counts.

### Results

Most patients had prediabetes (preDM) or diabetes mellitus (DM), 72.5% by oGTT, and 52.5% by HbA1c criteria. Pulse rates and insulin resistance indices (Homeostatic Model Assessment of Insulin Resistance, HOMA IR; Matsuda) were significantly different between patients classified as normal or with preDM/DM, using either oGTT or HbA1c criteria. Patients with preDM/DM by HbA1c but not oGTT criteria also had higher waist/hip ratios,

**Funding:** SUM: Pilot award from the University of Rochester Medical Center Department of Medicine. URL: https://www.urmc.rochester.edu/medicine.aspx The sponsor did not play a role in study design, data collection, analysis, decision to publish or preparation of the manuscript.

**Competing interests:** The authors have declared that no competing interests exist.

triglyceride, and CRP levels (p<0.05). A strong negative correlation was found between average daily step counts and CRP levels (rho = -0.62, p<0.001). Higher UKDDQ scores, or unhealthier diet habits, were associated with higher fasting plasma glucose (rho = 0.28, p = 0.08), triglyceride levels (rho = 0.31, p = 0.05), and insulin resistance (HOMA IR: rho = 0.31, p = 0.06). Higher PHQ-9 scores correlated with lower 2h-oGTT glucose levels (rho = -0.37, p<0.05).

## Conclusions

OGTT screening is superior to HbA1c screening in detecting preDM and DM early. Patients identified with preDM/DM by oGTT or HbA1c screening are insulin-resistant and have higher pulse rates. Abdominal obesity, unfavorable lipid profiles, and higher CRP levels were noted in patients screened by HbA1c, but not by oGTT. Low physical activity, low depression scores, and unhealthy diet habits were associated with higher CRP and higher glucose and triglyceride levels, respectively. Future studies should assess the impact of specifically tailored individual lifestyle counseling and medical management interventions in this high-risk population.

## Introduction

As of 2020, 14.2 million adults in the US were affected by serious mental illness [1]. People with serious mental illness (SMI), which includes schizophrenia, schizoaffective disorder, and bipolar disorder, die at much younger ages than people without SMI, most often from cardiovascular (CV) causes [2–4]. Proposed mediators of these adverse health outcomes are high rates of obesity, type 2 diabetes mellitus (T2DM), hypertension, smoking, and social deprivation [5, 6]. Psychotropic medications, and in particular second-generation antipsychotic medications (SGAs), have been directly implicated as mediators of adverse CV outcomes [6, 7]. Olanzapine and clozapine, for instance, while highly effective in controlling psychotic symptoms, often cause hyperphagia, weight gain, insulin resistance, and hyperglycemia, thereby promoting obesity and T2DM [8]. In addition, poor diet habits and sedentary behaviors may increase the risk for obesity, metabolic syndrome, and T2DM in these patients [9, 10], although not reported consistently [11].

Thus, it is prudent to screen for and treat diabetes and associated risk factors early and aggressively in people with SMI on SGAs to prevent adverse CV outcomes. Accordingly, guidelines recommend regular screening for diabetes mellitus and other cardiometabolic risk factors before and after starting antipsychotic medications [7, 12].

The importance of early and accurate diabetes detection in this high-risk population raises a practical question: How good are available screening tests? Screening by hemoglobin A1c (HbA1c) or fasting glucose alone can miss a substantial number of patients with prediabetes and diabetes compared to oral glucose tolerance testing (oGTT) [13, 14]. For instance, in a cohort of patients on olanzapine, screening by fasting glucose alone detected only one patient with diabetes; screening by oGTT detected seven patients [15]. In a larger study of patients on various antipsychotic medications, HbA1c screening missed most patients with diabetes (3 of 4) compared to oGTT screening [16]. Based on the above results and a review of the literature, Mitchell and co-authors proposed a two-step screening approach, starting with HbA1c measurements, followed by oGTT for patients with an HbA1c ≥ 5.7% [16].

Considering the aforementioned high risk for adverse CV outcomes, patients with SMI on SGAs should also be screened for additional cardiometabolic risk factors. Hyperglycemia, insulin resistance, and hypertriglyceridemia often coexist in patients with SMI in the presence of SGA use [15, 17]. Elevated C-reactive protein (CRP) levels are another well-recognized risk-enhancing factor for CV disease [18]. They have been associated with abdominal obesity [19] as well as diabetes [20]. Furthermore, elevated CRP levels have been noted in patients with schizophrenia [21], as well as in patients with depression [22]. Together, insulin resistance, hypertriglyceridemia, and elevated CRP levels likely contribute to an increased risk for T2DM and, more importantly, CV disease. Therefore, we reasoned that all the above should be considered when risk-stratifying patients with SMI on SGAs. We also postulated that lifestyle habits, mood, and cognition influence cardiometabolic risk profiles.

According to the above hypotheses, by way of a prospective, single-center cohort study, we set out to screen patients with SMI on SGAs for diabetes mellitus by oGTT and HbA1c. We also assessed insulin resistance, body mass indices (BMI), waist/hip ratios, comprehensive metabolic and lipid panels, and CRP levels. Lastly, we assessed the above patient population's lifestyle habits, mood, and cognition.

We aimed to test the following hypotheses: A) OGTT is more sensitive in diagnosing prediabetes or diabetes compared to HbA1c. B) A diagnosis of prediabetes or diabetes by oGTT and/or HbA1c is associated with insulin resistance, dyslipidemia, and low-grade inflammation. C) Poor lifestyle habits (diet and physical activity), lower cognition, and/or depressed mood may increase the risk of metabolic syndrome, prediabetes, and/or diabetes, as well as low-grade inflammation.

## Methods

### Patients

Patients were recruited at a single primary care setting for people with SMI at the University of Rochester Medical Center (URMC), New York, between 7/2019 and 7/2021.

Patients aged ≥ 18 and ≤ 75 years with a documented diagnosis of schizophrenia, schizoaffective, or bipolar disorder were included if they were on a stable dose of a second-generation antipsychotic medication (SGA; olanzapine, clozapine, quetiapine, risperidone, aripiprazole and/or paliperidone) for at least three months. Clinical diagnoses were made by a psychiatrist according to the Diagnostic and Statistical Manual of Mental Illnesses, fifth edition, and were extracted from the patient's electronic health record (EHR). Exclusion criteria were as follows: diagnosis of diabetes mellitus (documented HbA1c ≥ 6.5%, fasting glucose levels ≥ 126 mg/dL, random glucose levels ≥ 200 mg/dL or active therapy with antidiabetic medications such as metformin), pregnancy, or breast-feeding, treatment with glucocorticoids, active substance use <30 days before screening (except for nicotine), impaired hepatic function (liver transaminases >3 times upper normal limit), significantly impaired renal function (GFR <30 mL/min), active or history of pancreatic disease (acute or chronic pancreatitis and/or lipase/amylase >2 times upper normal), or heart failure (NYHA class III or IV).

### Procedures

Eligible patients, identified by a review of their electronic health records (EHR) by a research team member according to the above-listed inclusion and exclusion criteria, were approached by their primary care team and consented by a member of the research team if they were interested in participating. To ensure the patient's decisional verbal and written consent capacity, we also conducted a screening test according to the University of California, San Diego Brief Assessment of Capacity to Consent (UBACC). The UBACC used here included 9 questions

focusing on understanding and appreciation of the information concerning the research protocol. A score of 12 (out of 18) was required for participation in the study.

Screening procedures included a standard 75g oGTT and HbA1c. A diagnosis of prediabetes was defined as a fasting plasma glucose of 100–125 mg/dL, a 2h plasma glucose of 140–199 mg/dL, or a HbA1c of 5.7–6.4%. A diagnosis of diabetes mellitus was defined as a fasting plasma glucose of $\geq$ 126 mg/dL, a 2h plasma glucose of $\geq$ 200 mg/dL, or a HbA1c $\geq$ 6.5% [23]. We also obtained a comprehensive medical and medication history, height, and weight to calculate body mass index (BMI), measured waist and hip circumferences according to WHO guidelines [24], and blood pressure and pulse rates. Dietary habits were assessed using a 24-item food frequency questionnaire (UKDDQ = UK Diabetes and Diet Questionnaire). This questionnaire was specifically developed for clinicians to rapidly assess dietary habits in patients with or at risk for diabetes and provided excellent repeatability and acceptable agreement with reported food diaries [25]. Physical activity records were assessed as follows: upon completion of the above oGTT, each patient was given a pedometer (Omron HJ-320, Omron Healthcare Inc., FL) with instructions to wear it for 7 days before return. Daily step counts were entered for days 2–6 of each 7-day period.

Cognition was assessed via SLUMS (Saint Louis University Mental Status) examination. The above questionnaire was chosen as it allows for a rapid (<10 min/questionnaire) assessment of patients and can thus be administered during regular clinic visits. While not specifically studied in patients with SMI, the SLUMS examination has been shown to reliably detect mild cognitive impairment and dementia in veterans and various elderly populations [26–28]. Mood was assessed by completing the PHQ-9 (Patient Health Questionnaire-9), a similarly quick and reliable screening tool for depression across a broad spectrum of patients with mental illnesses [29].

On the day of the screening visit, patients arrived at the test center after an overnight fasting period (>8h). Fasting blood samples were obtained to determine comprehensive metabolic and lipid profile, CRP, plasma glucose, HbA1c, and insulin levels. Thereafter, patients were instructed to ingest a 75g-glucose drink (Trutol, Thermo Fisher Scientific Inc, Middletown, VA) within 10min. A second blood sample was obtained 120 min after the start of glucose ingestion to determine 2h plasma glucose, insulin, and CRP levels. Medical history was obtained, and measurements of anthropometrics, blood pressure, pulse rates, and above-noted surveys (UKDDQ, SLUMS, PHQ-9) were conducted before the second blood draw.

The URMC Research Subjects Review Board reviewed and approved the study protocol (original study ID RSRB00073753, follow-up study ID STUDY00002050).

## Laboratory measurements

Blood samples for plasma analytes were immediately placed on ice, and blood samples for serum analytes were allowed to clot at room temperature for 30 min. They were then spun at 3200 rpm for 5 min, followed by separation of serum and transport on ice to the laboratory or for subsequent storage at -80C.

Serum electrolytes, creatinine, albumin, alanine aminotransaminase (ALT), aspartate aminotransaminase (AST), alkaline phosphatase (AP), bilirubin, total cholesterol, triglycerides (TG), high-density lipoprotein cholesterol (HDL) and low-density lipoprotein cholesterol (LDL), plasma glucose and insulin were measured at the URMC's Clinical Laboratory Improvement Amendments(CLIA)-certified clinical laboratory and assessed using standard methods.

CRP levels were measured by ELISA (ab260058, Abcam Plc, Cambridge, UK).

HOMA IR and Matsuda indices were calculated as previously described [30, 31].

### Statistical analysis

SPSS version 28 was used for descriptive analyses and statistical comparisons between groups. Kolmogorov-Smirnov tests determined if continuous data were normally distributed (p>0.1). Based on Kolmogorov-Smirnov testing, it was determined that critical data (fi glucose, insulin, triglyceride levels) were not normally distributed. Thus, nonparametric tests (Kruskal Wallis, Mann Whitney U, Wilcoxon Signed Rank tests, Spearman correlations) were used to compare continuous variables. Accordingly, medians and interquartile ranges (IQR) are reported for descriptive analyses. Chi-square tests were applied to compare categorical variables. A p-value of <0.05 was considered statistically significant.

This study was designed to be exploratory. That said, according to previous work, the incidence of diabetes mellitus in patients on SGAs was expected to be between 10–20% [15, 16]. We aimed to screen 100 patients to have at least ten patients in each subgroup (normal, prediabetes, or diabetes), thus allowing for meaningful statistical comparisons of the hypothesized risk factors, including insulin resistance, lipid markers, CRP, lifestyle habits (diet and physical activity), cognition and mood.

## Results

Following the prescreening of potentially eligible patients by EHR (>550 patients), 210 patients were deemed eligible based on the above inclusion and exclusion criteria. Of those, 119 patients attended respective clinic visits and were approached by their primary care provider, 66 patients consented to participate, 40 completed screening, and their data were included in the final analyses. Twenty-six patients did not complete screening for the following reasons: ten patients were not interested anymore, three patients started metformin after being consented, six patients could not be reached for scheduling, three patients were not mentally stable for screening visits, two patients could not be screened due to an acute medical illness, one patient was switched to a non-eligible antipsychotic medication, and one patient died. Based on the less-than-intended sample size, we conducted binary subgroup analyses of hypothesized risk factors, comparing patients with normal results to patients with prediabetes or diabetes (n = 11-29/group).

Of the patients screened, most had schizophrenia (schizophrenia: n = 23, schizoaffective disorder: n = 10, bipolar disorder: n = 7), most were on olanzapine or clozapine (n = 32, 80%, olanzapine: n = 17, clozapine: n = 15). Over half of the patients were also on antidepressant medications (n = 23, 57.5%), and 27.5% were on mood stabilizers (n = 11). Most patients were male (n = 32, 80%) and of White non-Hispanic race/ethnicity (n = 30, 75%); 37.5% (n = 15) were active smokers, 10% (n = 4) reported occasional marijuana use (regular substance use was an exclusion criterium), only one patient reported daily alcohol use. Most patients (n = 21, 52.5%) had a BMI of at least 30 kg/m$^2$ (see Table 1).

### Hypothesis A: OGTT is more sensitive in diagnosing prediabetes or diabetes compared to HbA1c

Following diabetes screening, patients were classified as having normal results, prediabetes (preDM), or diabetes mellitus (DM), either by oGTT (fasting and/or 2h plasma glucose levels) or HbA1c criteria according to current ADA guidelines [32]. By HbA1c criteria, 19 patients were classified as normal, whereas 19 patients had preDM, and two patients had DM. By oGTT criteria, only 11 patients were classified as normal, 21 patients had preDM, and eight patients had DM (see Fig 1). In short, HbA1c screening alone classified more patients as normal compared to oGTT screening (42% more or 8 of 19) and missed 75% of patients with DM (6 of 8). Concordant classification by HbA1c and oGTT was noted in 14 patients with preDM/DM and 11 with normal results.

**Table 1. Demographic, anthropometric, psychometric, and lifestyle data for all patients and subgroups categorized by HbA1c.**

|  | All | HbA1c < 5.7% | HbA1c≥5.7% |
|---|---|---|---|
| Age (years) | 51.5 (22) | 48 (29) | 53 (17) |
| Gender (% male) | 80.0 | 78.9 | 81.0 |
| Ethnicity (% white) | 75.0 | 78.9 | 71.4 |
| Smoking (%) | 37.5 | 36.8 | 38.1 |
| BMI (kg/m$^2$) | 32.4 (8.4) | 28.0 (8.9) | 32.8 (9.6) |
| Waist/hip ratio | 1.05 (0.08) | 1.02 (0.09) | 1.08 (0.1)[#] |
| Systolic BP (mm Hg) | 130 (15) | 135 (22) | 130 (12) |
| Diastolic BP (mm/Hg) | 76 (15) | 74 (10) | 80 (15) |
| Pulse (rate/min) | 86.5 (28) | 74 (25) | 92 (23)[#] |
| UKDDQ diet score | 38 (15) | 33 (18) | 40 (15) |
| PHQ-9 depression score | 5 (8) | 6.5 (9) | 5 (7) |
| SLUMS cognition score | 22 (7) | 24 (10) | 21 (6) |
| Step count (average/day) | 3485 (3120) | 3681 (2435) | 3562 (3992) |

Data are shown as percentages or medians and interquartile ranges (IQR, in brackets). Patients were categorized by HbA1c (< 5.7% = normal, ≥ 5.7% = elevated).

[#] p < 0.05 for comparisons between subgroups.

## Hypothesis B: A diagnosis of prediabetes or diabetes by oGTT and/or HbA1c is associated with insulin resistance, dyslipidemia, and low-grade inflammation

We compared anthropometric and cardiometabolic variables between patients classified as normal or having preDM/DM, using either the classification by oGTT or HbA1c. Patients with preDM/DM by oGTT criteria had significantly higher pulse rates (p = 0.03), higher HOMA IR (p = 0.007), and lower Matsuda indices (p = 0.002) compared to patients classified as normal. Patients with preDM/DM by HbA1c criterium also had significantly higher pulse rates (p = 0.01), higher HOMA IR (p<0.001), and lower Matsuda indices (p = 0.002) compared to patients with normal results (see Fig 2). In addition, patients with preDM/DM by HbA1c, but not oGTT criteria, had higher waist circumferences (p = 0.008) and waist/hip ratios (p = 0.002), as well as higher triglyceride levels (p = 0.04) than patients classified as normal (see Fig 2A–2F). Furthermore, higher CRP levels (p = 0.04) were noted in patients with preDM/DM by HbA1c (see Fig 2A–2F), but not oGTT criteria (not shown). Taken together, patients with preDM/DM classified by oGTT or HbA1c were more insulin resistant and had higher pulse rates than patients without preDM/DM. However, patients with preDM/DM classified by HbA1c but not oGTT also had abdominal obesity, relative hypertriglyceridemia, and low-grade inflammation compared to patients without preDM/DM.

Of note, BMIs did not differ significantly in patients with or without preDM/DM by either oGTT or HbA1c criteria, suggesting that BMI may not be as well-suited to identify patients at risk for prediabetes or diabetes.

## Hypothesis C: Poor lifestyle habits (diet and physical activity), lower cognition, and/or depressed mood may increase the risk of metabolic syndrome, prediabetes, and/or diabetes, as well as low-grade inflammation

We assessed measures of mood (PHQ-9 score), cognition (SLUMS score), and lifestyle habits (UKDDQ scores, i.e., diet habits; average 3d step count) in relation to cardiometabolic variables, specifically to fasting and 2h glucose, HbA1c, insulin resistance indices (HOMA IR,

## Diagnosis by HbA1c:

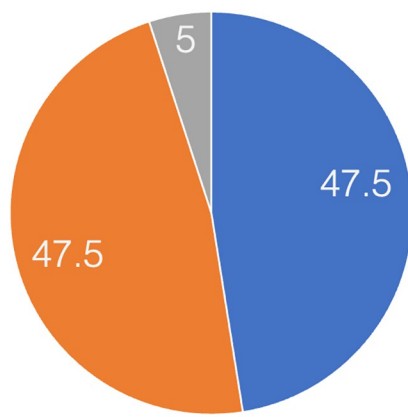

## Diagnosis by oGTT:

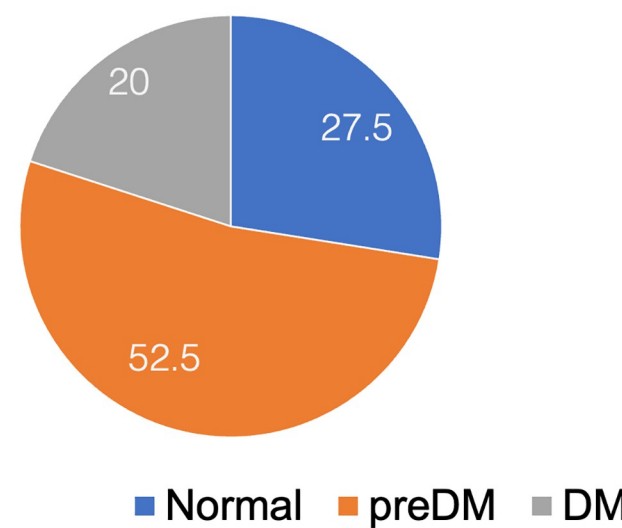

■ Normal ■ preDM ■ DM

**Fig 1. Percentages of patients classified as normal, with prediabetes (preDM) or diabetes mellitus (DM) by oGTT or HbA1c.**

Matsuda indices), triglyceride, HDL, LDL, and/or CRP levels (see Spearman correlations in Table 2).

Firstly, we noted a positive correlation between depression and cognition scores (Spearman's rho = 0.37, p = 0.02), indicating that patients with higher cognition scores were more likely to be depressed than patients with lower cognition scores in our patient population. No correlation was noted between depression, cognition scores, diet habits (UKDDQ), or physical activity (average step count). Neither cognition nor mood was associated with fasting glucose levels, HbA1c, insulin resistance indices (HOMA IR, Matsuda indices), triglyceride, HDL, LDL, and/or CRP levels. However, patients with lower PHQ-9 scores had higher 2h-glucose levels during oGTT (Spearman's rho = -0.37, p = 0.02).

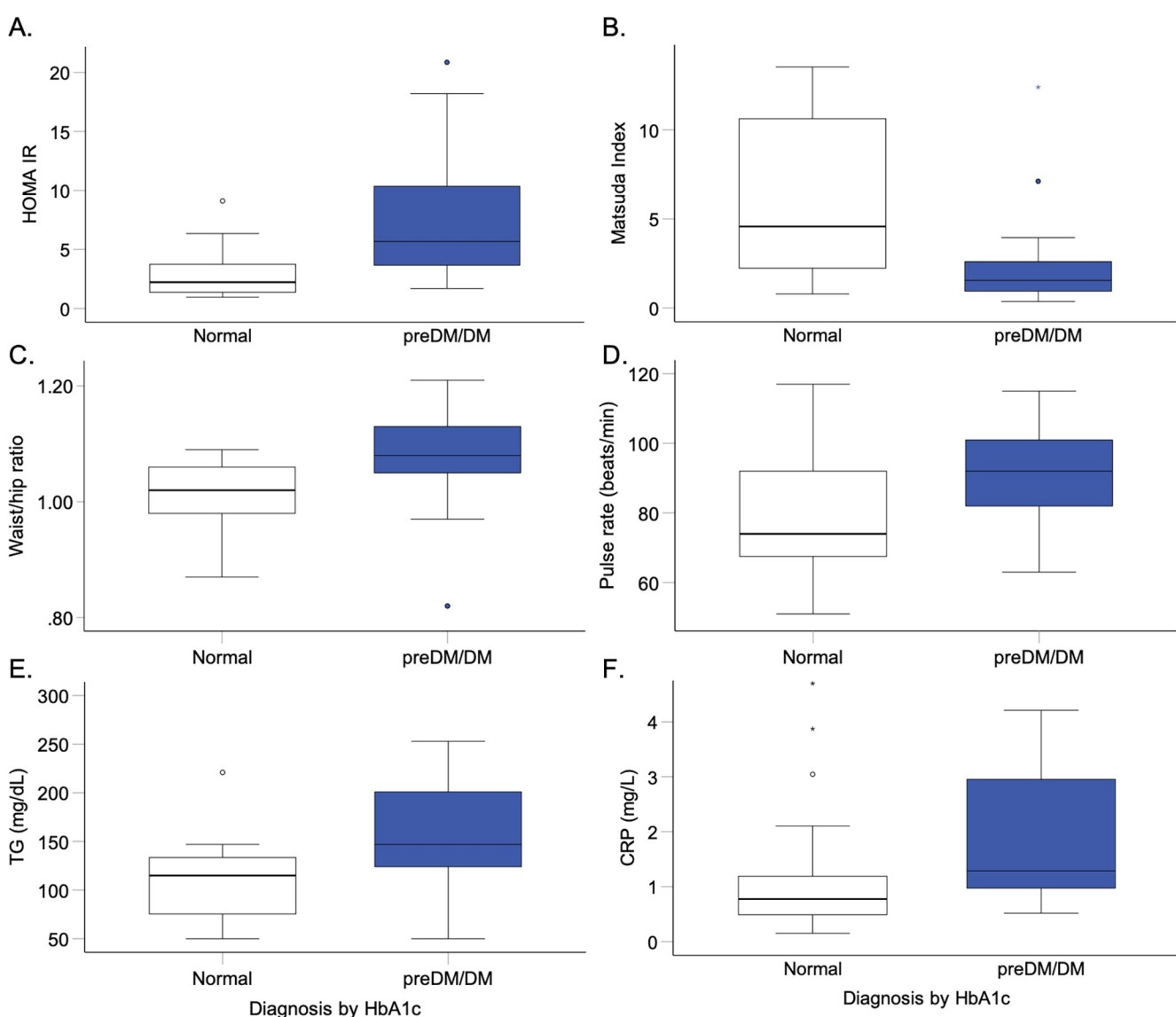

**Fig 2.** Boxplot comparisons of HOMA IR (A, p<0.001), Matsuda indices (B, p = 0.002), waist/hip ratios (C, p = 0.002), pulse rates (D, p = 0.01), triglyceride (TG, E, p = 0.04) and CRP levels (F, p = 0.04) between patients classified as normal (white) or having prediabetes or diabetes (preDM/DM, blue) by HbA1c.

Secondly, we assessed patients' reported diet habits in relation to glucose and lipid metabolism. The UKDDQ is a 24-item food-frequency questionnaire that was specifically designed for people with or at risk for diabetes mellitus and to be easy to use in day-to-day healthcare settings [25]. This questionnaire can be completed within 5–10 min, demonstrating excellent test-retest reliability and good agreement with food diaries [25]. In our patient population, higher UKDDQ scores (diets low in fiber, fruits, and vegetables and high in unhealthy carbohydrates and fats) were associated with higher fasting glucose (Spearman's rho = 0.28, p = 0.08) and triglyceride levels (Spearman's rho = 0.31, p = 0.05) as well as insulin resistance (HOMA IR, Spearman's rho = 0.31, p = 0.06). To our knowledge, this is the first study that applied the above food frequency questionnaire to patients with SMI. The above-reported trends between self-reported diet habits, fasting glucose, triglyceride levels, and insulin resistance suggest that the UKDDQ screen can provide meaningful insights into the diet habits of patients with SMI.

**Table 2. Spearman correlations between psychometric (mood, cognition scores) and lifestyle habit data (diet questionnaire, average steps/d).**

|  | PHQ-9 | SLUMS | UKDDQ | Avg steps/d |
|---|---|---|---|---|
| HbA1c | -0.03 | -0.17 | 0.18 | 0.18 |
| Glucose 0min | 0.005 | 0.13 | 0.28* | -0.02 |
| Glucose120min | -0.37** | -0.28 | -0.17 | 0.11 |
| HOMA IR | -0.06 | 0.13 | 0.31* | -0.27 |
| Matsuda Index | 0.23 | 0.08 | -0.2 | 0.11 |
| Waist/hip ratio | -0.02 | -0.25 | 0.18 | 0.03 |
| TG | 0.06 | -0.12 | 0.31* | -0.23 |
| CRP | 0.1 | 0.06 | 0.11 | -0.62*** |

Mood evaluation: Patient Health Questionnaire = PHQ-9.

Cognition evaluation: Saint Louis University Mental Status examination = SLUMS.

Diet evaluation: UK Diabetes Diet Questionnaire = UKDDQ.

Physical activity evaluation: Average, avg, steps/d.

*p<0.1

**p<0.05

***p<0.001.

Last, we asked whether physical activity, defined as the average daily step count, affected cardiometabolic markers. Physical activity did not significantly correlate with glucose, triglyceride levels, or insulin resistance indices in our patient population. However, the average step count did correlate with BMI (Spearman's rho = -0.37, p = 0.04), waist circumference (Spearman's rho -0.46, p = 0.009), and CRP levels (Spearman's rho = -0.62, p<0.001, see Table 2). In other words, less active people were more likely to be obese; they had abdominal obesity and higher CRP levels.

## Discussion

People with SMI are at a high risk for CV morbidity and mortality [2–4]. Obesity, metabolic syndrome as well as type 2 diabetes mellitus shape this adverse CV risk profile and are promoted by the use of SGAs [6, 7]. Between 2003 and 2005, obesity rates were as high as 57% in patients on clozapine at a URMC psychiatric clinic patient population with SMI [33]. The prevalence of DM (documented diagnosis by medical record review or by screening of fasting blood glucose levels) was 14.2% in patients on various SGAs and as high as 25.7% in patients on clozapine; rates noted at least twice as high as in the general population at that time [34, 35].

In our current study, which was conducted about 15 years later in a similar patient population, 52.5% of all patients were obese. The rate of DM was 20% by screening with oGTT and 5% by screening with HbA1c. While our obesity rates were comparable to those of the previous studies, DM rates were lower in this study. Both results are surprising considering the global trends of obesity and diabetes and the fact that our current patient population was over ten years older than the patients described previously ([34, 35]: average age 41 years, current study: average age 51 years). One would expect higher rates of obesity and diabetes since 2004, but also in older patients [36, 37]. The differences in diabetes rates are explained by the patient selection: the previous studies included primarily patients with a documented diagnosis and thus mainly reflect diabetes prevalence. Our study included only patients without a previous diagnosis of diabetes and thus reported only DM incidence.

As hypothesized, we find that screening by oGTT detected more patients with either preDM or DM than screening by HbA1c. Specifically, by HbA1c criteria, 52.5% of patients

were classified as having preDM or DM, whereas 72.5% of patients had preDM or DM by oGTT criteria (of note, 93.3% of patients on clozapine had preDM or DM). By HbA1c screening, only 5% of patients had DM. OGTT screening, on the other hand, identified 20% of patients with DM. In other words, HbA1c screening missed most patients with DM and about 25% of patients with preDM or DM compared to oGTT screening. Our reported rates of preDM and DM in people with SMI on antipsychotic medications confirm the results of others in the United States and Europe: by oGTT screening, others identified up to 75% of patients with preDM, and up to 20% of patients with DM with higher rates noted by oGTT as compared to HbA1c screening, similar to our results [15–17]. In summary, screening by oGTT is more sensitive in detecting preDM or DM in people with SMI, as reported in the general population [13, 16]. That said, HbA1c is a much more appealing diabetes screening test. HbA1c testing does not require fasting and can be easily obtained during a regular clinic visit by point-of-care testing. Notably, when using a cut-off of 5.7%, HbA1c is highly predictive of a future diagnosis of DM [38]. Furthermore, elevated HbA1c levels have been associated with an increased risk of adverse CV outcomes [39, 40].

We thus asked whether screening by oGTT or HbA1c would better identify people with adverse cardiometabolic profiles (such as abdominal adiposity, insulin resistance, elevated blood pressure or pulse rates, abnormal lipid profiles, or higher CRP levels). We find that insulin resistance (higher HOMA IR, lower Matsuda indices) and higher pulse rates characterized patients with preDM/DM identified by oGTT or HbA1c criteria. In addition, patients with preDM/DM identified by HbA1c but not by oGTT also had significantly higher waist circumferences, waist-to-hip ratios, triglyceride, and CRP levels. In a European study of people with SMI on SGAs, insulin resistance, higher waist circumferences, and triglyceride levels were noted in association with a diagnosis of preDM by oGTT; HbA1c levels were not reported [17].

Based on all of the above data, we conclude that either screening by HbA1c, using a cutoff of 5.7%, or screening by oGTT appear to be well-suited to identify patients with unfavorable cardiometabolic risk profiles early and should be checked in patients with SMI before and regularly after starting antipsychotic medications according to previously published recommendations [7, 16]. Since oGTT is more sensitive in identifying patients with preDM/DM, we recommend performing an oGTT in patients with a normal HbA1c but with clinical evidence of insulin resistance or metabolic syndrome (acanthosis nigricans and/or abdominal adiposity). The above approach will allow for an early diagnosis and, more importantly, the implementation of therapy for T2DM, which is key to preventing cardiometabolic complications.

In our quest to characterize patients with adverse cardiometabolic risk profiles, the observed higher pulse rates in our patients with preDM/DM compared to patients with normal results is worth noting. Higher resting heart rates have been linked to obesity [41], insulin resistance [42], diabetes risk [43], and chronic inflammation [44], as well as CV morbidity and mortality by others [45–47]. The above and our results suggest that elevated heart/pulse rates should be considered a risk-modifying cardiometabolic biomarker. Dysautonomia, specifically a heightened adrenergic tone, has been discussed as the underlying mechanism of elevated heart/pulse rates [43]. In conclusion, abdominal obesity, insulin resistance, elevated CRP levels, hyperglycemia, hypertriglyceridemia, and higher heart/pulse rates should all be considered potential harbingers of adverse cardiometabolic outcomes. All should be included in a patient's cardiometabolic risk assessment, and more importantly, all should be taken into account when making treatment decisions, such as when or if to start metformin, an incretin analog, and/or a sodium-glucose-transporter-2 inhibitor.

Our final hypothesis was that mood, cognition, and lifestyle habits would shape the cardiometabolic risk profiles in our patients. We find that cognition and depression scores did not

significantly affect most cardiometabolic risk markers recorded here, with the exception that patients with low depression scores had higher 2h-glucose levels. This result seems counterintuitive as distress related to depression should result in higher, not lower, glucose levels as shown for the general population, and thus needs further exploration in people with SMI [48, 49]. Our assessments of daily physical activity and diet habits revealed potentially meaningful trends: (A) patients with lower daily step counts had higher CRP levels and (B), patients with poor diet habits were more insulin-resistant and had higher fasting glucose and triglyceride levels.

Higher CRP levels have been associated with a sedentary lifestyle in the general population [50]. More importantly, higher CRP levels have been associated with increased CV morbidity and mortality [51, 52]. In people with SMI, low physical activity is common and associated with male gender, being single or unemployed, having fewer years of education, a higher body mass index, longer mental illness duration, lower cardiorespiratory fitness, a diagnosis of schizophrenia or taking antidepressant and antipsychotic medications [10]. One study of people with psychosis found that self-reported low physical activity was associated with higher CRP levels [53]. Our data extend the above observations by providing objective evidence that lower physical activity is closely linked to higher CRP levels in people with SMI on SGAs. We also confirm that lower physical activity is associated with a higher BMI and waist circumference. The above findings underscore the interconnectedness of physical activity, inflammation, and obesity. They also stress the urgent need to address sedentary lifestyle behaviors in people with SMI on SGAs to improve their overall health.

Finally, our study explored the utility of a short food frequency questionnaire (UKDDQ) in assessing diet habits in patients with SMI on SGAs. Compared to previously reported UKDDQ scores obtained in people with or at risk for DM in the UK (average HbA1c 6.9% and 5.7%, respectively), we find considerably higher scores in our patients with SMI (>10 points higher), signifying that our patients were eating a rather unhealthy diet (lower in fiber, fruits, vegetables, and higher in unhealthy carbohydrates and fats) [54]. Notably, higher UKDDQ scores, or, in other words, self-reported, unhealthier diet habits in our patients, were associated with higher fasting glucose and triglyceride levels. The latter findings validate the information obtained in this brief 24-item questionnaire and suggest that the above survey can provide meaningful information in patients with SMI.

The following limitations of our study need to be considered. The cross-sectional design precludes conclusions related to temporal trends or causality of our observations. Moreover, despite screening a large patient population in a primary care setting for people with mental illnesses, the number of patients who were eligible, interested, consented, and who completed all screening procedures was relatively small, which limits the statistical power of subgroup comparisons. There were several reasons: a significant number of patients already had DM, were on metformin for preDM, or had other significant co-morbidities precluding study participation. Other patients were either not interested in diabetes screening or unwilling to complete a >2h screening visit that included repeated blood draws and completion of several questionnaires. All of the above could have introduced a potential selection bias to our study population, which may have shaped our results. That said, the above observations also point to opportunities for health education for people with SMI to raise awareness of the adverse risks associated with metabolic syndrome and diabetes mellitus. Lastly, compared to 2h-oGTT rapid point-of-care HbA1c testing will likely be more accepted for diabetes screening, especially by people with SMI.

In summary, our study confirms that oGTT screening is superior to HbA1c screening in detecting DM early. Secondly, we demonstrate that patients identified with preDM/DM by oGTT or HbA1c screening are insulin-resistant and have higher pulse rates. Abdominal

obesity, unfavorable lipid profiles, and evidence of chronic inflammation (higher CRP levels) were noted in patients screened by HbA1c but not by oGTT. Finally, we show that the above adverse cardiometabolic risk profile is at least in part promoted by a sedentary lifestyle and poor diet habits.

Future studies should assess the value of comprehensive cardiometabolic screening measures such as the above in predicting diabetes and adverse CV events in patients with SMI over time. Finally, our results emphasize the urgent need for tailored individual lifestyle counseling and medical management interventions in this high-risk patient population.

## Supporting information

**S1 Data. The complete raw data set for all 40 patients is attached (prospective-data-set-revised-5-2023).**
(XLSX)

## Acknowledgments

The authors wish to thank Dr. Stephen R Hammes for his critical advice and review of the manuscript.

## Author Contributions

**Conceptualization:** Susanne U. Miedlich, Telva E. Olivares, J. Steven Lamberti.

**Data curation:** Susanne U. Miedlich, Priya Sahay, Lauren Bainbridge.

**Formal analysis:** Susanne U. Miedlich.

**Funding acquisition:** Susanne U. Miedlich.

**Investigation:** Susanne U. Miedlich, Priya Sahay, Lauren Bainbridge.

**Methodology:** Susanne U. Miedlich, Telva E. Olivares, J. Steven Lamberti, Kavaljit H. Chhabra.

**Project administration:** Susanne U. Miedlich, Priya Sahay, Kavaljit H. Chhabra, Lauren Bainbridge.

**Resources:** Susanne U. Miedlich, J. Steven Lamberti, Kevin P. Brazill, Kavaljit H. Chhabra.

**Supervision:** Susanne U. Miedlich.

**Validation:** Susanne U. Miedlich.

**Visualization:** Susanne U. Miedlich.

**Writing – original draft:** Susanne U. Miedlich.

**Writing – review & editing:** Susanne U. Miedlich, Priya Sahay, Telva E. Olivares, J. Steven Lamberti, Diane S. Morse, Kevin P. Brazill, Kavaljit H. Chhabra.

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
