## [Decision Letter · Decision Letter 0]

9 Apr 2024

PONE-D-23-34690Lifestyle and mood correlates of cardiometabolic risk in people with serious mental illness on second-generation antipsychotic medications.PLOS ONE

Dear Dr. Miedlich,

Thank you for submitting your manuscript to PLOS ONE. After careful consideration, we feel that it has merit but does not fully meet PLOS ONE’s publication criteria as it currently stands. Therefore, we invite you to submit a revised version of the manuscript that addresses the points raised during the review process.

**ACADEMIC EDITOR: **

We look forward to receiving your revised manuscript.

Kind regards,

Hidetaka Hamasaki

Academic Editor

PLOS ONE

Reviewers' comments:

Reviewer's Responses to Questions

**Comments to the Author**

1. Is the manuscript technically sound, and do the data support the conclusions?

Reviewer #1: Yes

Reviewer #2: Yes

2. Has the statistical analysis been performed appropriately and rigorously? 

Reviewer #1: Yes

Reviewer #2: I Don't Know

3. Have the authors made all data underlying the findings in their manuscript fully available?

Reviewer #1: Yes

Reviewer #2: Yes

4. Is the manuscript presented in an intelligible fashion and written in standard English?

Reviewer #1: Yes

Reviewer #2: Yes

5. Review Comments to the Author

Reviewer #1: I'm delighted and so glad to review this very good work. It was my pleasure to offer insights and feedback on lifestyle and mood correlates of cardiometabolic risk in people with serious mental illness on second-generation antipsychotic medications.

Abstract and introduction were very good in writing and formative.

methodology

sampling technique and recruitment process were not clear to me

number of patients and how was the calculation of sample size?

in results section

Write a footnote for abbreviations, tests of significance, and degrees of significance under the tables.

make an outline for limitation and conclusions.

cheers

thanks again

Reviewer #2: It is an interesting topics, "Life style and mood correlates of cardio metabolic risk in people with serious mental illness on SGA medications". I have some concerns regarding your manuscript.

1. Methods: your participants were recruited at a single primary care setting for people with SMI, but it is not clear that how you recruit the participants/Justify your sampling techniques?

2. Reference citation: you have to cite each references in each sentences/paragraphs before a dot.

3 what is/are the limitation of your study? Please include it in your manuscript.

6. PLOS authors have the option to publish the peer review history of their article (what does this mean?). If published, this will include your full peer review and any attached files.

Reviewer #1: **Yes: **Ayman Mohamed El-Ashry

Reviewer #2: **Yes: **Agmas Wassie Abate

---

## [Author Response · Author response to Decision Letter 0]

2 Jun 2024

Response to reviewer #1: 

1. Methodology: Sampling technique and recruitment process were not clear to me,

number of patients, and how was the calculation of sample size?

We appreciate the question. We added more detail to procedures, statistical analysis and results sections to explain our approach to recruitment and sample size calculations. 

2. Results section: Write a footnote under the tables for abbreviations, tests of significance, and degrees of significance. Make an outline for limitations and conclusions.

We added the details requested above to the bottom of each table. We chose to detail limitations (lines 447-458) and a summary of the conclusions (lines 464-470) at the end of the discussion, thus omitting the above in the results section to avoid repetitive content. 

Reviewer #2: 

1. Methods: Your participants were recruited at a single primary care setting for people with SMI, but it is not clear that how you recruited the participants. Justify your sampling techniques.

Thank you, we added more descriptive detail to the procedures and results sections.

2. Reference citation: You have to cite each reference in each sentence/paragraph before a dot.

Thank you, revisions were made according to your suggestions.

3. What is/are the limitation(s) of your study? Please include it in your manuscript.

As mentioned above, we chose to detail the limitations of our study in the discussion (lines 447-458).

---

## [Decision Letter · Decision Letter 1]

25 Jun 2024

Lifestyle and mood correlates of cardiometabolic risk in people with serious mental illness on second-generation antipsychotic medications.

PONE-D-23-34690R1

Dear Dr. Miedlich,

We’re pleased to inform you that your manuscript has been judged scientifically suitable for publication and will be formally accepted for publication once it meets all outstanding technical requirements.

Kind regards,

Hidetaka Hamasaki

Academic Editor

PLOS ONE

Additional Editor Comments (optional):

Thank you for revising your manuscript. Although Reviewer #1 has not reviewed the revised version of your manuscript; however, the editor has confirmed that you have adequately responded to the reviewer's comments.

Reviewers' comments:

Reviewer's Responses to Questions

**Comments to the Author**

1. If the authors have adequately addressed your comments raised in a previous round of review and you feel that this manuscript is now acceptable for publication, you may indicate that here to bypass the “Comments to the Author” section, enter your conflict of interest statement in the “Confidential to Editor” section, and submit your "Accept" recommendation.

Reviewer #2: All comments have been addressed

2. Is the manuscript technically sound, and do the data support the conclusions?

Reviewer #2: Yes

3. Has the statistical analysis been performed appropriately and rigorously? 

Reviewer #2: Yes

4. Have the authors made all data underlying the findings in their manuscript fully available?

Reviewer #2: Yes

5. Is the manuscript presented in an intelligible fashion and written in standard English?

Reviewer #2: Yes

6. Review Comments to the Author

Reviewer #2: (No Response)

7. PLOS authors have the option to publish the peer review history of their article (what does this mean?). If published, this will include your full peer review and any attached files.

Reviewer #2: **Yes: **Agmas Wassie Abate

---

## [Editor Report · Acceptance letter]

2 Jul 2024

PONE-D-23-34690R1 

PLOS ONE

Dear Dr. Miedlich, 

I'm pleased to inform you that your manuscript has been deemed suitable for publication in PLOS ONE. Congratulations! Your manuscript is now being handed over to our production team.

Kind regards, 

on behalf of

Dr. Hidetaka Hamasaki 

Academic Editor

PLOS ONE